# On Algebraic Properties of Primitive Eisenstein Integers with Applications in Coding Theory

**DOI:** 10.3390/e27040337

**Published:** 2025-03-24

**Authors:** Abdul Hadi, Uha Isnaini, Indah Emilia Wijayanti, Martianus Frederic Ezerman

**Affiliations:** 1Department of Mathematics, Universitas Gadjah Mada, Sekip Utara Bulaksumur 21, Yogyakarta 55281, Indonesia or abdulhadi@lecturer.unri.ac.id (A.H.); isnainiuha@ugm.ac.id (U.I.); ind_wijayanti@ugm.ac.id (I.E.W.); 2Department of Mathematics, Universitas Riau, Tampan, Pekanbaru 28293, Indonesia; 3School of Physical and Mathematical Sciences, Nanyang Technological University, 21 Nanyang Link, Singapore 637371, Singapore

**Keywords:** Eisenstein integers, unit group, set partitioning, signal constellation

## Abstract

An even Eisenstein integer is a multiple of an Eisenstein prime of the least norm. Otherwise, an Eisenstein integer is called odd. An Eisenstein integer that is not an integer multiple of another one is said to be primitive. Such integers can be used to construct signal constellations and complex-valued codes over Eisenstein integers via a carefully designed modulo function. In this work, we establish algebraic properties of even, odd, and primitive Eisenstein integers. We investigate conditions for the set of all units in a given quotient ring of Eisenstein integers to form a cyclic group. We perform set partitioning based on the multiplicative group of the set. This generalizes the known partitioning of size a prime number congruent to 1 modulo 3 based on the multiplicative group of the Eisenstein field in the literature.

## 1. Introduction

Eisenstein integers, named after the mathematician Ferdinand Gotthold Max Eisenstein, are complex numbers that can be expressed as α:=a+bρ, where *a* and *b* are integers and $\rho =e^{2\pi i/3}=\frac{-1}{2}+\frac{\sqrt{3}}{2}i$ such that $\rho^{3}=1$ in C and $i^{2}=-1$. The integers *a* and *b* are the *real part* and the *rho part*, respectively. Since the set of all Eisenstein integers, denoted by Z[ρ], forms a commutative ring with identity, it is commonly referred to as *the ring of Eisenstein integers* [1]. Occasionally, it is also called *the ring of Eisenstein–Jacobi integers*. The integers possess remarkable geometric properties. They form a hexagonal lattice in the complex plane, making them particularly useful in coding theory, cryptography, and signal processing. They allow for optimal packing and minimal energy configurations in various practical setups. The ring Z[ρ] is a Euclidean domain and, hence, is also a principal ideal domain and a unique factorization domain. Inspired by the algebraic properties of Gaussian integers discussed in, e.g., [2,3], many researchers have discovered properties of Z[ρ] by generalizing important properties of the ring of integers Z and the ring of Gaussian integers Z[i]. We know of fundamental concepts such as the factor ring, the unit structure of the factor ring, and the Euler-Totient function on Eisenstein integers from results presented in [4,5,6,7].

Gullerud and Mbirika in [7] introduced the notion of even and odd numbers in Z[ρ]. Revisiting their motivation, the prime number 2 has the least norm, in this case defined as the absolute value, in Z. The quotient by the ideal generated by the even prime 2 has two cosets that partition Z into even and odd integers. Since 1−ρ and its *associates* are primes with the least norm, to be formally defined below, in Z[ρ], we can pick 1−ρ to play the role of an even prime in Z[ρ], just like 2 in Z. Unlike in Z, however, the quotient by the ideal generated by 1−ρ is the set whose elements partition Z[ρ] into three sets, which we call *even*, *odd of Type 1*, and *odd of Type 2* sets. Some of their properties were investigated based on the norm and the sum of the real and the rho parts in [7].

In Z[ρ], an Eisenstein integer that is not an integer multiple of another is called *primitive*. Such an integer can be used to construct signal constellations and complex-valued codes over Eisenstein integers. These codes are obtained through a modulo function. Complex-valued codes are mathematical representations of coded symbols in communication systems, where codewords are constructed from complex numbers rather than real-valued symbols. These codes are particularly useful in digital communication for efficient modulation and error correction. We have provided a necessary and sufficient condition for an Eisenstein integer to be primitive in [8]. In that same work, we also constructed signal constellations for codes over Z[ρ] by studying primitive and non-primitive Eisenstein integers. In communication systems, a *signal constellation* is a physical diagram that depicts all possible symbols used by a signaling system to transmit data better. Mathematically, a signal constellation is a set of the residual class rings obtained by taking some modulo. Eisenstein integers have been used in designing denser and more efficient patterns in signal transmission. Such patterns have been shown to be beneficial in modern approaches, such as multiple input multiple output (MIMO) in [9], physical-layer network coding in [10,11,12,13], and compute and forward in [14].

Primitive Eisenstein integers exhibit excellent algebraic and number theoretic properties for applications in cryptography and error-correcting codes. There is an isomorphism between Z[ρ] modulo a primitive Eisenstein integer and Z modulo an integer, based on Theorem 8 below. In this work, we focus on discovering further algebraic properties of primitive Eisenstein integers as well as even and odd Eisenstein integers.

The multiplicative group of units in the quotient ring of Eisenstein integers has applications in coding theory. It has been used as QAM signals in [15,16], for enhanced spatial modulation in [17], and as a tool for set partitioning and multilevel-coded modulation in [18]. The set partitioning method leverages on the cyclic group structure of the units in the Eisenstein field Z[ρ]/〈ψ〉 such that the norm of ψ is a prime integer q≡1(mod3).

Constructions of codes over a number of other rings based on their primitive elements have been proposed in the literature. They utilize an isomorphism between a quotient ring induced by a primitive element and the ring of integers modulo the norm of a primitive element. The isomorphism sends a one-dimensional signal to a higher-dimensional signal. This general approach has been successfully performed to obtain codes. Examples include codes over Z[i] built based on primitive Gaussian integers in [19], codes over Lipschitz integers based on primitive Lipschitz integers in [20], and codes over Hurwitz integers, again using the primitive Lipschitz integers in [21,22,23]. The properties of primitive Lipschitz integers that are beneficial for encoding can be found in [24].

Li, Gan, and Ling in [25] provided a necessary and sufficient condition for two Eisenstein integers to be relatively prime.

**Theorem** **1**([25]). *Two arbitrary Eisenstein integers α and θ are relatively prime if and only if*gcdNρ(α),Nρ(θ),23Im(αθ¯)=1,
*with $\;\bar{\;\theta \;}\;$ being the conjugate of θ, or, equivalently,*
gcdNρ(α),Nρ(θ),Re(αθ¯)−13Im(αθ¯)=1.

We also know, this time from [26] that, if a Gaussian integer α and its conjugate $\;\bar{\;\alpha \;}\;$ are relatively prime, then $\alpha^{-1}\;\left(\mathrm{mod}\;\;\bar{\;\alpha \;}\;\right)$ is an integer. This fact is useful in constructing multi-channel modulo samplers from Gaussian integers. It seems that no one has checked if the analogue of the fact and its application work over Z[ρ].

Freudenberger and Shavgulidze in [18] considered finite sets of Eisenstein integers $E_{\eta }=\left\{\mu_{\eta }\left(\alpha \right)\;:\;\alpha \in Z_{N_{\rho }\left(\eta \right)}\right\}$. They paid special attention to the case of η=ψ, which is a primitive and prime Eisenstein integer whose norm is a prime q≡1(mod3), as a two-dimensional signal constellation. Computing $\mu_{\psi }\left(\alpha \right)$ according to (Equation 2) below, the set of all units in $E_{\eta }$, denoted by ${\left(E_{\psi }\right)}^{*}$, can then be considered as a signal constellation for the general spatial modulation. In general, $E_{\eta }$ is a representation of the quotient ring of Eisenstein integers only when η is primitive. In such a case, we can then partition ${\left(E_{\psi }\right)}^{*}$ into $n=\frac{\varphi (\psi )}{6}$ subsets, indexed by j∈{0,1,…,n−1}, as$${\left(E_{\psi }\right)}_{\left(j\right)}^{*}=\left\{\alpha^{n+j},\alpha^{2n+j},\alpha^{3n+j},\alpha^{4n+j},\alpha^{5n+j},\alpha^{6n+j}\right\}=\left\{\pm \alpha^{j},\pm \rho \alpha^{j},\pm \left(1+\rho \right)\alpha^{j}\right\},$$
with α being a generator of the cyclic group ${\left(E_{\psi }\right)}^{*}$ that corresponds to the generator of the cyclic group ${\left(Z\left[\rho \right]/\langle \psi \rangle \right)}^{*}$. We can perform set partitioning on ${\left(E_{\psi }\right)}_{\left(j\right)}^{*}$ according to the following theorem to obtain a larger minimum distance in each subset.

**Theorem** **2**(Proposition 1 in [18]). *Let j∈{0,1,…,n−1}. The minimum Euclidean distance in ${\left(E_{\psi }\right)}_{\left(j\right)}^{*}$ is $∥\alpha^{j}∥$. We can partition ${\left(E_{\psi }\right)}_{\left(j\right)}^{*}$ further into three subsets*$${\left(E_{\psi }\right)}_{\left(j\right)}^{*}=\left\{\pm \alpha^{j}\right\}\cup \left\{\pm \rho \alpha^{j}\right\}\cup \left\{\pm \left(1+\rho \right)\alpha^{j}\right\},$$
*each with minimum Euclidean distance $2∥\alpha^{j}∥$. We can also partition ${\left(E_{\psi }\right)}_{\left(j\right)}^{*}$ into two subsets*
$${\left(E_{\psi }\right)}_{\left(j\right)}^{*}=\left\{\alpha^{j},\rho \alpha^{j},-\left(1+\rho \right)\alpha^{j}\right\}\cup \left\{-\alpha^{j},-\rho \alpha^{j},\left(1+\rho \right)\alpha^{j}\right\},$$
*each with minimum Euclidean distance $\sqrt{3}∥\alpha^{j}∥$.*

In this paper, we gladly report the following contributions.

1.We establish further algebraic properties of primitive, even, and odd Eisenstein integers. We then answer Question 6.1 in [7]. Let ψ be an Eisenstein prime such that $N_{\rho }\left(\psi \right)=q$ is a prime integer and q≡1(mod3).a.Are the (non-associate) distinct pairs of primes ψ and $\;\bar{\;\psi \;}\;$ always of the same odd class? The answer is *yes, they are*.b.Does the corresponding *q* predict the odd class of ψ and $\;\bar{\;\psi \;}\;$? The answer is *no, it does not*.2.Taking advantage of Theorem 1, our Theorem 22 confirms that, if Eisenstein integers α and $\;\bar{\;\alpha \;}\;$ are relatively prime, then $\alpha^{-1}\;\left(\mathrm{mod}\;\;\bar{\;\alpha \;}\;\right)$ is in Z. This result leads to a construction of multi-channel modulo samplers.3.We prove important properties of the set of all units in a quotient ring of Z[ρ] when the set forms a cyclic group. The multiplicative group of the set leads to a nice set partitioning that generalizes Theorem 2 by using the modulo function in (Equation 1), which differs from the original modulo function in (Equation 2).

In terms of organization, Section 2 reviews known properties of Eisenstein integers. Section 3 presents our new results. We establish the algebraic properties of Eisenstein integers related to their being even, odd, or primitive. We look into the cyclic groups in the quotient ring. Set partitioning based on the multiplicative group of units in the quotient ring is the focus of Section 4. Section 5 highlights the role of primitive Eisenstein integers in the relevant code constructions. Section 6 contains a summary and several concluding remarks.

## 2. Preliminaries

This section recalls known properties of Eisenstein integers related to their being prime, primitive, odd or even. We also recall useful results on the quotient rings and and the unit group in a quotient ring.

### 2.1. Ring of Eisenstein Integers

Since $\rho =\frac{-1}{2}+\frac{\sqrt{3}}{2}i$ is a complex primitive third root of unity, we have $\rho^{3}=1$ and $\left(\rho -1\right)\left(\rho^{2}+\rho +1\right)=0$ implies $\rho^{2}+\rho +1=0$. Addition and multiplication in Z[ρ] are defined, respectively, by$$\begin{array}{cc} \left(a+b\rho )+(c+d\rho \right) & =(a+c)+(b+d)\rho \\ \left(a+b\rho )\cdot (c+d\rho \right) & =(ac-bd)+(ad+bc-bd)\rho . \end{array}$$
The *conjugate* and *norm* of α=a+bρ∈Z[ρ] for a,b∈Z are defined, respectively, as$$\;\bar{\;\alpha \;}\;=\left(a-b\right)-b\rho \;\mathrm{and}\;N_{\rho }\left(\alpha \right)=N_{\rho }\left(\;\bar{\;\alpha \;}\;\right)=\alpha \;\;\bar{\;\alpha \;}\;=a^{2}+b^{2}-ab\in Z.$$
By definition, $N_{\rho }\left(\alpha \right)={∥\alpha ∥}^{2}$, where ∥·∥ denotes the Euclidean distance, and the norm is multiplicative since $N_{\rho }\left(\alpha \theta \right)=N_{\rho }\left(\alpha \right)\;N_{\rho }\left(\theta \right)$ for all α,θ∈Z[ρ].

The division algorithm works over Z[ρ], i.e., for α,η≠0∈Z[ρ], there exists a unique quotient θ and a remainder δ in Z[ρ] such that α=θη+δ and $N_{\rho }\left(\delta \right)<N_{\rho }\left(\eta \right)$. Since Z[ρ] is a Euclidean domain (ED), it is a principal ideal domain (PID) and a unique factorization domain (UFD).

In Z[ρ], an element η divides α, denoted by η∣α, if there exists θ∈Z[ρ] such that α=θη. We say that α is a *unit* in Z[ρ] if αλ=1 for some λ∈Z[ρ]. A unit has a unique multiplicative inverse. It is known that α is a unit if and only if $N_{\rho }\left(\alpha \right)=1$ and that Z[ρ] has 6 units. These are ±1,±ρ, and ±(1+ρ). We say that α and β are *associates*, denoted by α∼η, if α=θη for some unit θ∈Z[ρ]. The associates of α=a+bρ are ±α, ±ρα, and ±(1+ρ)α, with ρα=−b+(a−b)ρ and (1+ρ)α=(a−b)+aρ.

The *greatest common divisor* (GCD) ω of α,θ∈Z[ρ], denoted by ω:=gcd(α,θ), is the largest Eisenstein integer in terms of modulus, up to multiplication by any unit, that divides *both*α and θ. Every common divisor of α and θ divides ω.

Let Q(·) denote the quantization to the closest Eisenstein integer in as [27,28]. Fixing a nonzero η∈Z[ρ], we can define a *modulo function*$\mu_{\eta }\left(\cdot \right)$ as(1)$$\mu_{\eta }\left(\alpha \right):=\alpha \;\left(\mathrm{mod}\;\eta \right)=\alpha -Q\left(\frac{\alpha }{\eta }\right)\cdot \eta .$$
Algorithm 1, which computes a remainder $\mu_{\eta }\left(\alpha \right)$ when α is divided by η, is a slight adaptation of the version in [11,27].

We highlight that the modulo function $\mu_{\eta }$ in (Equation 1) is different from the modulo function(2)$$\mu_{\eta }\left(\alpha \right)=\alpha -\left\lfloor \frac{\alpha }{\eta }\right\rceil \eta ,$$
with ⌊·⌉ denoting the rounding to the nearest integer as defined in [29]. For avoidance of doubt, we choose to define ⌊x⌉:=⌊x+0.5⌋ for all x∈R in this paper. Our choice is somewhat arbitrary. If so desired, one can define ⌊x⌉:=⌈x−0.5⌉ for all x∈R.
**Algorithm 1** Finding a remainder δ:=α(modη) on input a given α and a fixed η.z←αη=Re(z)+Im(z)i and z−ρ←Re(z−ρ)+Im(z−ρ)i.The nearest Eisenstein integers $\theta_{1}\in Z\left[\sqrt{3}i\right]$ and $\theta_{2}\in \rho +Z\left[\sqrt{3}i\right]$ areθ1←⌊Re(z)⌉+Im(z)33iθ2←⌊Re(z−ρ)⌉+Im(z−ρ)33i+ρ.$\delta_{1}\leftarrow \alpha -\theta_{1}\;\eta$ and $\delta_{2}\leftarrow \alpha -\theta_{2}\;\eta$.Output δ:=α(modη) based onδ←δ1, if Nρ(δ1)<Nρ(δ2), or Nρ(δ1)=Nρ(δ2)andRe(θ1)<Re(θ2),δ←δ2, otherwise.

We use the modulo function in (Equation 1) because it gives us $N_{\rho }\left(\mu_{\eta }\left(\alpha \right)\right)=N_{\rho }\left(\delta \right)\leq N_{\rho }\left(\alpha \right)$ for every α∈Z[ρ]. In contrast, using (Equation 2) over Z[ρ] implies the existence of η∈Z[ρ] such that $N_{\rho }\left(\mu_{\eta }\left(\alpha \right)\right)>N_{\rho }\left(\alpha \right)$ for some α∈Z[ρ].

**Example** **1.**
*Let η=−6+5ρ and α=5. Since*

$$\frac{5}{-6+5\rho }=\frac{5(-11-5\rho )}{\left(-6+5\rho )(-11-5\rho \right)}=\frac{-55}{91}+\frac{-25}{91}\rho ,$$

*we have*

$$\begin{array}{c} \left\lfloor \frac{5}{-6+5\rho }\right\rceil =\left\lfloor \frac{-55}{91}\right\rceil +\left\lfloor \frac{-25}{91}\right\rceil \rho =-1+0\rho =-1. \end{array}$$

*Applying (Equation 2), we obtain*

$$\begin{array}{cc} \mu_{\eta }\left(5\right) & =5-(-1)(-6+5\rho )=-1+5\rho \;and\\ N_{\rho }\left(\mu_{\eta }\left(5\right)\right) & =N_{\rho }\left(-1+5\rho \right)=31>25=N_{\rho }\left(5\right). \end{array}$$



### 2.2. Prime and Primitive Eisenstein Integers

An α∈Z[ρ] is called *(Eisenstein) prime* if α cannot be expressed as α=θη where θ and η are not units in Z[ρ]. In other words, α is Eisenstein prime if *all* of its divisors are of the form uα with u∈{±1,±ρ,±(1+ρ)}. Otherwise, α is *(Eisenstein) composite*. An α=a+bρ is primitive if gcd(a,b)=1.

Eisenstein primes are classified as follows:1.The prime 1−ρ and its associates.2.The prime c+dρ, with $N_{\rho }\left(c+d\;\rho \right)=q$ such that *q* is a prime in Z, with q≡1(mod3), and its associates.3.The prime p∈Z such that p≡2(mod3) and its associates.

For the rest of this paper, let β:=1−ρ and let *p* and *q* be prime integers such that p≡2(mod3) and $q=\psi \;\;\bar{\;\psi \;}\;\equiv 1\;\left(\mathrm{mod}\;3\right)$, where ψ and $\;\bar{\;\psi \;}\;$ are non-associate Eisenstein primes. We denote a generic Eisenstein prime by γ.

**Remark** **1.***Units as well as Eisenstein primes β and ψ up to associates are primitive Eisenstein integers. Any prime integer p≡2(mod3) and its associates are *not* primitive Eisenstein integers. We note that 5+4ρ is primitive but not an Eisenstein prime since 5+4ρ=(1−ρ)(2+3ρ).*

**Theorem** **3**([30]). *If $\gamma_{1}$ and $\gamma_{2}$ are Eisenstein primes such that $N_{\rho }\left(\gamma_{1}\right)=N_{\rho }\left(\gamma_{2}\right)$, then $\gamma_{1}∼\gamma_{2}$ or $\gamma_{1}∼\;\bar{\;\gamma_{2}\;}\;$. If $N_{\rho }\left(\gamma_{1}\right)=3$, then $\gamma_{1}∼\beta$. If $N_{\rho }\left(\gamma_{1}\right)=p^{2}$, with p≡2(mod3), then $\gamma_{1}∼\;\bar{\;\gamma_{1}\;}\;$. Lastly, if q is a prime integer such that $N_{\rho }\left(\gamma_{1}\right)=q\equiv 1\;\left(\mathrm{mod}\;3\right)$, then $\gamma_{1}≁\;\bar{\;\gamma_{1}\;}\;$.*

**Theorem** **4**([8]). *Given any two elements α,θ∈Z[ρ], we have $N_{\rho }\left(\alpha \right)=N_{\rho }\left(\theta \right)\in Z$ if and only if α∼θ or $\alpha ∼\;\bar{\;\theta \;}\;$.*

Gullerud and Mbirika stated in Theorem 5.8 of [7] that any power of an Eisenstein prime ψ is a primitive element. To prove this valid claim, they had assumed that if the norms of two Eisenstein integers are the same, then they are associates. This *assumption is invalid*. Theorem 4 states that it does *not* hold in general. We reproduce the original theorem and supply a proof in Appendix A. Our proof uses Theorem 4.

**Theorem** **5**(Theorem 5.8 in [7]). *Let ψ=x+yρ be a prime in Z[ρ]. If $N_{\rho }\left(\psi \right)=q\equiv 1\;\left(\mathrm{mod}\;3\right)$ be such that q is a prime in Z, then $\psi^{n}$ is a primitive Eisenstein integer for all n∈N.*

**Proof.** See Appendix A. □

In another recent work, we have established a necessary and sufficient condition for an Eisenstein integer to be primitive.

**Theorem** **6**([8]). *An Eisenstein integer η is primitive if and only if $\eta ∼\beta^{r}\psi_{1}^{r_{1}}\cdots \psi_{m}^{r_{m}}$, with*
*r∈{0,1}, m, and $r_{i}$ are nonnegative integers,**$N_{\rho }\left(\psi_{i}\right)=q_{i}\in Z$ is a prime such that $q_{i}\equiv 1\;\left(\mathrm{mod}\;3\right)$ for 0≤i≤m,**$q_{i}\neq q_{j}$ for i,j∈{0,1,…,m} such that i≠j.*

### 2.3. On the Quotient Ring of Eisenstein Integers

Since Z[ρ] is a PID, any ideal is of the form 〈η〉 for some η∈Z[ρ]. A congruence in Z[ρ] modulo 〈η〉 can then be defined. For any α,θ∈Z[ρ], we have α≡θ(modη) if and only if α−θ∈〈η〉. For any α∈Z[ρ], the *equivalence class* of α with respect to η, denoted by ${\left[\alpha \right]}_{\eta }$, is defined to be$${\left[\alpha \right]}_{\eta }=\left\{\theta \in Z\left[\rho \right]\;:\;\theta \equiv \alpha \;\left(\mathrm{mod}\;\eta \right)\right\}.$$
The set $\left\{{\left[\alpha \right]}_{\eta }\;:\;\alpha \in Z\left[\rho \right]\right\}$ forms the *quotient ring*Z[ρ]/〈η〉.

We will soon make use of three results from [4].

**Theorem** **7**([4]). *If η∈Z[ρ]∖{0} is such that η=a+bρ=t(m+nρ), with gcd(a,b)=t and gcd(m,n)=1, then the complete residue system is*$$Z\left[\rho \right]/\langle \eta \rangle =\{{\left[x+y\rho \right]}_{\eta }\;:\;0\leq x<tN_{\rho }\left(m+n\rho \right),\;0\leq y<t\},$$
*with ${\left[x+y\rho \right]}_{\eta }:=x+y\rho +\langle \eta \rangle$.*

**Theorem** **8**([4]). *If η is a primitive Eisenstein integer, then $Z\left[\rho \right]/\langle \eta \rangle ≅Z_{N_{\rho }\left(\eta \right)}$.*

**Theorem** **9**([4]). *If n∈N, then $Z\left[\rho \right]/\langle n\rangle ≅Z_{n}\left[\rho \right]$.*

The ring $Z_{n}\left[\rho \right]$ is known as *the ring of Eisenstein integers modulo n*.

### 2.4. Even and Odd Eisenstein Integers

By Theorem 7, for β=1−ρ∈Z[ρ], we have $Z\left[\rho \right]/\langle \beta \rangle =\{{\left[0\right]}_{\beta },\;{\left[1\right]}_{\beta },\;{\left[2\right]}_{\beta }\}$, with$$\begin{array}{cc} {\left[0\right]}_{\beta } & =\{x+y\rho \in Z[\rho ]\;:\;x+y\rho \equiv 0\;(\mathrm{mod}\;\beta )\},\\ {\left[1\right]}_{\beta } & =\{x+y\rho \in Z[\rho ]\;:\;x+y\rho \equiv 1\;(\mathrm{mod}\;\beta )\},\\ {\left[2\right]}_{\beta } & =\{x+y\rho \in Z[\rho ]\;:\;x+y\rho \equiv 2\;(\mathrm{mod}\;\beta )\}. \end{array}$$
An Eisenstein integer α is *even* if $\alpha \in {\left[0\right]}_{\beta }$. An Eisenstein integer α is *odd* if α is in ${\left[1\right]}_{\beta }\cup {\left[2\right]}_{\beta }$. More precisely, α is *odd of Type-1* if $\alpha \in {\left[1\right]}_{\beta }$. It is *odd of Type-2* if $\alpha \in {\left[2\right]}_{\beta }$. We denote the respective sets of all even, odd Type-1, and odd Type-2 Eisenstein integers by *E*, $O_{1}$, and $O_{2}$.

**Remark** **2.**
*By Theorem 6, an Eisenstein integer of the form ${\left(1+\rho \right)}^{\ell }\beta \psi_{1}^{r_{1}}\cdots \psi_{m}^{r_{m}}$ is even primitive and an Eisenstein integer of the form ${\left(1+\rho \right)}^{\ell }\psi_{1}^{r_{1}}\cdots \psi_{m}^{r_{m}}$ is odd primitive.*


We have a simple characterization based on the sum of the real and the rho parts.

**Theorem** **10**([7]). *For any x+yρ∈Z[ρ], we have*
*i.* *x+yρ∈E if and only if x+y≡0(mod3) if and only if $N_{\rho }\left(x+y\rho \right)\equiv 0\;\left(\mathrm{mod}\;3\right)$.**ii.* *$x+y\rho \in O_{1}$ if and only if x+y≡1(mod3), which implies $N_{\rho }\left(x+y\rho \right)\equiv 1\;\left(\mathrm{mod}\;3\right)$.**iii.* *$x+y\rho \in O_{2}$ if and only if x+y≡2(mod3), which implies $N_{\rho }\left(x+y\rho \right)\equiv 1\;\left(\mathrm{mod}\;3\right)$.*

**Example** **2.**
*A prime β, its associates and multiples are even Eisenstein integers. The other primes are odd Eisenstein integers. The prime $\psi_{1}=2+3\rho$ is an odd Eisenstein integer of Type-2. The prime $\psi_{2}=3+4\rho$ is an odd Eisenstein integer of Type-1. Any prime integer p≡2(mod3) is an odd Eisenstein integer of Type-2.*


**Theorem** **11**([7]). *If α, θ, τ, $\tau^{\prime }$, σ, and $\sigma^{\prime }$ are in Z[ρ] such that θ∈E, $\tau ,\tau^{\prime }\in O_{1}$, and $\sigma ,\sigma^{\prime }\in O_{2}$, then*$$\alpha \cdot \theta \in E,\;\tau \cdot \sigma \in O_{2},\;\tau \cdot \tau^{\prime }and\;\sigma \cdot \sigma^{\prime }\in O_{1}.$$

### 2.5. Unit Group in the Quotient Ring of Eisenstein Integers

The set of all units in Z[ρ]/〈η〉, formally defined to be$${\left(Z[\rho ]/\langle \eta \rangle )\right)}^{*}=\left\{{\left[\alpha \right]}_{\eta }\in Z\left[\rho \right]/\langle \eta \rangle \;:\;\gcd\left(\alpha ,\eta \right)=1\right\},$$
is a group under multiplication. The Euler-Totient function with respect to η∈Z[ρ] is the order of unit group ${\left(Z[\rho ]/\langle \eta \rangle \right)}^{*}$,$$\varphi_{\rho }\left(\eta \right)=\left|{\left(Z[\rho ]/\langle \eta \rangle \right)}^{*}\right|.$$
If η and 1 are associates, then $\varphi_{\rho }\left(\eta \right)=1$.

Recall that γ denotes a generic Eisenstein prime. We have the following easy way to determine the units in $Z\left[\rho \right]/\langle \gamma^{n}\rangle$.

**Theorem** **12**([4]). *The set of all units in $Z\left[\rho \right]/\langle \gamma^{n}\rangle$ are*$$\begin{array}{cc} {\left(Z\left[\rho \right]/\langle \beta^{n}\rangle \right)}^{*} & =\{{\left[x+y\rho \right]}_{\beta^{n}}\in Z\left[\rho \right]/\langle \beta^{n}\rangle \;:\;x+y≢0\;\left(\mathrm{mod}\;3\right)\},\\ {\left(Z\left[\rho \right]/\langle \psi^{n}\rangle \right)}^{*} & =\{{\left[x\right]}_{\psi^{n}}\in Z\left[\rho \right]/\langle \psi^{n}\rangle \;:\;\gcd\left(x,q\right)=1\},\\ {\left(Z\left[\rho \right]/\langle p^{n}\rangle \right)}^{*} & =\{{\left[x+y\rho \right]}_{p^{n}}\in Z\left[\rho \right]/\langle p^{n}\rangle \;:\;\gcd\left(x,p\right)=1\;or\gcd\left(y,p\right)=1\}. \end{array}$$

The unit group $Z_{n}^{*}$ in Z is cyclic if and only if $n\in \{2,4,p^{k},2p^{k}\}$, where *p* is an odd prime and *k* is a positive integer. A necessary and sufficient condition for the unit group ${\left(Z\left[\rho \right]/\langle \eta \rangle \right)}^{*}$ to be cyclic is known.

**Theorem** **13**([31,32]). *A unit group ${\left(Z\left[\rho \right]/\langle \eta \rangle \right)}^{*}$ is cyclic if and only if*$$\eta \;is\;an\;element\;or\;an\;associate\;of\;an\;element\;in\;\{\beta ,\beta^{2},2\beta ,\psi^{k},p\},$$
*where k∈N, ψ is an Eisenstein prime such that $N_{\rho }\left(\psi \right)=q\equiv 1\;\left(\mathrm{mod}\;3\right)$, and p is a prime integer such that p≡2(mod3).*

**Theorem** **14**([7]). *If η∈Z[ρ]∖{0}, then $\varphi_{\rho }\left(\eta \right)$ is even, except when η is a unit, or η and 2 are associates.*

**Theorem** **15**([33]). *Let η∈Z[ρ]∖{0} be such that η is not a unit. If β≁η≁2, then $6|\varphi_{\rho }\left(\eta \right)$.*

**Theorem** **16**([33]). *If η∼n for an n∈Z, then $\varphi \left(n\right)|\varphi_{\rho }\left(\eta \right)$ and $\varphi \left(\varphi \left(n\right)\right)\leq \varphi \left(\varphi_{\rho }\left(\eta \right)\right)$. In particular, for any positive integer k,*$$\varphi_{\rho }\left(\eta \right)=\left\{\begin{array}{cc} \varphi (n), & ifn=1,\\ n\varphi (n), & ifn=3^{k},\\ {\left(\varphi \left(n\right)\right)}^{2}, & ifn=q^{k},\;with\;q\equiv 1\;\left(\mathrm{mod}\;3\right)\;being\;a\;prime\;integer,\\ \left(n+\frac{n}{p}\right)\;\varphi \left(n\right), & ifn=p^{k},\;with\;p\equiv 2\;\left(\mathrm{mod}\;3\right)\;being\;a\;prime\;integer. \end{array}\right.$$

## 3. Further Properties of Eisenstein Integers

We discuss further properties of primitive, even, and odd Eisenstein integers in the first subsection. The second subsection centers on the cyclic group of units in the quotient rings of Eisenstein integers.

### 3.1. On Even, Odd, and Primitive Eisenstein Integers

**Theorem** **17.**
*Let α,θ∈Z[ρ]. The following statements hold:*
*i.* 
*If α,θ∈E, then α+θ∈E.*
*ii.* 
*If $\alpha ,\theta \in O_{1}$, then $\alpha +\theta \in O_{2}$.*
*iii.* 
*If $\alpha ,\theta \in O_{2}$, then $\alpha +\theta \in O_{1}$.*
*iv.* 
*If $\alpha \in O_{1}$ and $\theta \in O_{2}$, then α+θ∈E.*
*v.* 
*If α∈E and $\theta \in O_{1}$, then $\alpha +\theta \in O_{1}$.*
*vi.* 
*If α∈E and $\theta \in O_{2}$, then $\alpha +\theta \in O_{2}$.*



**Proof.** A straightforward application of Theorem 10 confirms the assertions. □

**Theorem** **18.**
*Let α,θ∈Z[ρ].*
*i.* 
*If α∈E and α∼θ, then $\theta ,\;\bar{\;\alpha \;}\;\in E$.*
*ii.* 
*If $\alpha \in O_{1}$ and α∼θ, then θ=α, $\rho \alpha ,-\left(1+\rho \right)\alpha \in O_{1}$, and $-\theta \in O_{2}$.*
*iii.* 
*If $\alpha \in O_{2}$ and α∼θ, then $\theta =\alpha ,\rho \alpha ,-\left(1+\rho \right)\alpha \in O_{2}$, and $-\theta \in O_{1}$.*
*iv.* 
*If $\alpha \in O_{1}$, then $\;\bar{\;\alpha \;}\;\in O_{1}$.*
*v.* 
*If $\alpha \in O_{2}$, then $\;\bar{\;\alpha \;}\;\in O_{2}$.*



**Proof.** We proceed by items as listed.
i.Let α=a+bρ∈E and θ∼α. By Theorems 4 and 10, $N_{\rho }\left(\alpha \right)\equiv 0\;\left(\mathrm{mod}\;3\right)$ and $N_{\rho }\left(\theta \right)=N_{\rho }\left(\alpha \right)=N_{\rho }\left(\;\bar{\;\alpha \;}\;\right)\equiv 0\;\left(\mathrm{mod}\;3\right)$, affirming $\theta ,\;\bar{\;\alpha \;}\;\in E$.ii.Assuming $\alpha =a+b\rho \in O_{1}$ and α∼θ, Theorem 10 yields (a+b)≡1(mod3). Hence,$$\begin{array}{cc} -\alpha & =-a-b\equiv 2(a+b)\equiv 2\;(\mathrm{mod}\;3),\\ \rho \alpha & =-b+(a-b)\equiv a-2b\equiv a+b\equiv 1\;(\mathrm{mod}\;3),\\ -\rho \alpha & =b+(b-a)\equiv 2b-a\equiv 2b+2a\equiv 2(a+b)\equiv 2\;(\mathrm{mod}\;3),\\ -(1+\rho )\alpha & =(-a+b)-a\equiv b-2a\equiv b+a\equiv 1\;(\mathrm{mod}\;3),\\ (1+\rho )\alpha & =a-b+a\equiv 2a-b\equiv 2(a+b)\equiv 2\;(\mathrm{mod}\;3). \end{array}$$Thus, $\theta \in O_{1}$ and $-\theta \in O_{2}$ whenever θ∈{α,ρα,−(1+ρ)α}.iii.Assuming $\alpha =a+b\rho \in O_{2}$ and α∼θ, Theorem 10 yields (a+b)≡2(mod3). Hence,$$\begin{array}{cc} -\alpha & =-a-b\equiv 2(a+b)\equiv 4\equiv 1\;(\mathrm{mod}\;3),\\ \rho \alpha & =-b+(a-b)\equiv a-2b\equiv a+b\equiv 2\;(\mathrm{mod}\;3),\\ -\rho \alpha & =b+(b-a)\equiv 2b-a\equiv 2b+2a\equiv 2(a+b)\equiv 4\equiv 1\;(\mathrm{mod}\;3),\\ -(1+\rho )\alpha & =(-a+b)-a\equiv b-2a\equiv b+a\equiv 2\;(\mathrm{mod}\;3),\\ (1+\rho )\alpha & =a-b+a\equiv 2a-b\equiv 2(a+b)\equiv 4\equiv 1\;(\mathrm{mod}\;3). \end{array}$$Thus, $\theta \in O_{2}$ and $-\theta \in O_{1}$ whenever θ∈{α,ρα,−(1+ρ)α}.iv.Assuming $\alpha =a+b\rho \in O_{1}$, Theorem 10 gives us a+b≡1(mod3). Hence, a−b−b≡a−2b≡a+b≡1(mod3), ensuring $\;\bar{\;\alpha \;}\;\in O_{1}$v.Assuming $\alpha =a+b\rho \in O_{2}$, we obtain a+b≡2(mod3) by Theorem 10. Hence, a−b−b≡a−2b≡a+b≡2(mod3) and −a−b≡2(a+b)≡4≡1(mod3), which means $\;\bar{\;\alpha \;}\;\in O_{2}$.□

We can now answer Question 6.1 in [7].

By Theorem 18 iv. and v., we conclude that distinct primes ψ and $\;\bar{\;\psi \;}\;$ which are non-associates always belong to the the same odd class. Both are in $O_{1}$ or both are in $O_{2}$.Any prime q≡1(mod3) is always in $O_{1}$. By Theorem 11, however, both ψ and $\;\bar{\;\psi \;}\;$ are in $O_{1}$ or both are in $O_{2}$. We note, for example, that both $\psi_{1}=2+3\rho$ and $\;\bar{\;\psi_{1}\;}\;=-1-3\rho$ are in $O_{2}$. Both $\psi_{2}=3+\rho$ and $\;\bar{\;\psi_{2}\;}\;=2-\rho$ are in $O_{1}$, with $q=N_{\rho }\left(\psi_{1}\right)=N_{\rho }\left(\psi_{2}\right)=7\equiv 1\;\left(\mathrm{mod}\;3\right)$ being in $O_{1}$. Without further investigation, the *q* that corresponds to a given ψ does not automatically identify which odd class both ψ and $\;\bar{\;\psi \;}\;$ belong to.

The next result is a corollary to Theorem 11.

**Corollary** **1.**
*Given an odd Eisenstein integer*

$$\eta =\prod_{\psi_{i}\in O_{1}}\psi_{i}^{r_{i}}\prod_{\psi_{j}\in O_{2}}\psi_{j}^{s_{j}}\prod_{p_{k}\in O_{2}}p_{k}^{t_{k}},$$

*if $\left(\sum s_{j}+\sum t_{k}\right)\equiv 0\;\left(\mathrm{mod}\;2\right)$, then $\eta \in O_{1}$. Otherwise, $\eta \in O_{2}$.*


**Proof.** By Theorem 11, if $\left(\sum s_{j}+\sum t_{k}\right)\equiv 0\;\left(\mathrm{mod}\;2\right)$, then$$\prod_{\psi_{i}\in O_{1}}\psi_{i}^{r_{i}}\in O_{1}\;\mathrm{and}\prod_{\psi_{j}\in O_{2}}\psi_{j}^{s_{j}}\prod_{p_{k}\in O_{2}}p_{k}^{t_{k}}\in O_{1},$$
ensuring $\eta \in O_{1}$. If $\left(\sum s_{j}+\sum t_{k}\right)\equiv 1\;\left(\mathrm{mod}\;2\right)$, then $\prod_{\psi_{j}\in O_{2}}$$\psi_{j}^{s_{j}}\prod_{p_{k}\in O_{2}}p_{k}^{t_{k}}\in O_{2}$. Since $\prod_{\psi_{i}\in O_{1}}\psi_{i}^{r_{i}}\in O_{1}$, we confirm that $\eta \in O_{2}$. □

**Theorem** **19.**
*The associates and conjugates of a primitive Eisenstein integer are also primitive Eisenstein integers.*


**Proof.** If η=a+bρ such that gcd(a,b)=1, thengcd(a−b,−b)=gcd(b−a,b)=gcd(−a,−b)=gcd(a−b,a)=gcd(b−a,−a)=1.
Hence, its conjugate $\;\bar{\;\eta \;}\;=a-b-b\rho$ and associates ±α,±ρα and ±(1+ρ)α, with ρα=−b+(a−b)ρ and (1+ρ)α=(a−b)+aρ, are primitives. □

We know from Corollary 3 in [34] that an Eisenstein integer α=a+bρ and its conjugate $\;\bar{\;\alpha \;}\;$ are relatively prime if and only if gcd(a−b,b)=1 and gcd(a−2b,3)=1. Since gcd(a−b,b)=1 is equivalent to gcd(a,b)=1, and gcd(a−2b,3)=1 is equivalent to a+b≡±1(mod3), we can use the following equivalent expression of the corollary.

**Proposition** **1.**
*An Eisenstein integer α=a+bρ and its conjugate $\;\bar{\;\alpha \;}\;$ are relatively prime if and only if gcd(a,b)=1 and a+b≡±1(mod3). In short, an Eisenstein integer and its conjugate are relatively prime if and only if the Eisenstein integer is odd and primitive.*


The next result is a direct consequence of Proposition 1.

**Corollary** **2.**
*If an odd primitive Eisenstein integer η is not a unit, then η and $\;\bar{\;\eta \;}\;$ are not associates.*


**Proof.** Let η be an odd primitive Eisenstein integer such that η is not a unit. If η and $\;\bar{\;\eta \;}\;$ are associates, then $\gcd(\eta ,\;\bar{\;\eta \;}\;)=\eta$, which contradicts Proposition 1. □

**Theorem** **20.**
*Let $\eta =\prod_{\psi_{i}\in O_{1}}\psi_{i}^{r_{i}}\prod_{\psi_{j}\in O_{2}}\psi_{j}^{s_{j}}$ be an odd primitive Eisenstein integer. If $\sum s_{j}\equiv 0\;\left(\mathrm{mod}\;2\right)$, then $\eta \in O_{1}$. Otherwise, $\eta \in O_{2}$.*


**Proof.** By Theorem 11, if $\sum s_{j}\equiv 0\;\left(\mathrm{mod}\;2\right)$, then$$\prod_{\psi_{i}\in O_{1}}\psi_{i}^{r_{i}}\in O_{1}\;\mathrm{and}\prod_{\psi_{j}\in O_{2}}\psi_{j}^{s_{j}}\in O_{1},$$
implying $\eta \in O_{1}$. On the other hand, if $\sum s_{j}\equiv 1\;\left(\mathrm{mod}\;2\right)$, then $\prod_{\psi_{j}\in O_{2}}\psi_{j}^{s_{j}}\in O_{2}$. Since $\prod_{\psi_{j}\in O_{1}}\psi_{i}^{r_{i}}\in O_{1}$, it is clear that $\eta \in O_{2}$. □

**Theorem** **21.**
*Let η be a non-unit primitive Eisenstein integer.*
*i.* 
*If η is even, then $\gcd(\eta ,\;\bar{\;\eta \;}\;)=\beta$.*
*ii.* 
*If η and β are not associates, then η and $\;\bar{\;\eta \;}\;$ are also not associates.*



**Proof.** We prove the assertions according to their order of appearance.
i.Let u∈Z[ρ] be a unit and let $\eta =u\beta \psi_{1}^{r_{1}}\cdots \psi_{k}^{r_{k}}$. Since $\;\bar{\;\beta \;}\;=\left(1+\rho \right)\beta$, we have$$\;\bar{\;\eta \;}\;=\left(1+\rho \right)u\beta \;\bar{\;\psi_{1}^{r_{1}}\;}\;\cdots \;\bar{\;\psi_{k}^{r_{k}}\;}\;.$$By Proposition 1, we have $\gcd(\psi_{1}^{r_{1}}\cdots \psi_{k}^{r_{k}},\;\bar{\;\psi_{1}^{r_{1}}\;}\;\cdots \;\bar{\;\psi_{k}^{r_{k}}\;}\;)=1$. Thus, $\gcd(\eta ,\;\bar{\;\eta \;}\;)=\beta$.ii.For a contradiction, let us assume that η and $\;\bar{\;\eta \;}\;$ are associates. Let u∈Z[ρ] be a unit such that $\eta =u\psi_{1}^{r_{1}}\cdots \psi_{k}^{r_{k}}$. Then,$$\begin{array}{cc} \psi_{1}^{r_{1}}\cdots \psi_{k}^{r_{k}} & ∼\;{\bar{\;\psi_{1}\;}}^{r_{1}}\cdots \;{\bar{\;\psi_{k}\;}}^{r_{k}},\;\mathrm{contradicting Corollary}\;2. \end{array}$$If $\eta =u\beta \psi_{1}^{r_{1}}\cdots \psi_{k}^{r_{k}}$ for some unit u∈Z[ρ], then$$\begin{array}{cc} \beta \psi_{1}^{r_{1}}\cdots \psi_{k}^{r_{k}} & ∼\;\bar{\;\beta \;}\;\;{\bar{\;\psi_{1}\;}}^{r_{1}}\cdots \;{\bar{\;\psi_{k}\;}}^{r_{k}},\\ \beta \psi_{1}^{r_{1}}\cdots \psi_{k}^{r_{k}} & ∼\left(1+\rho \right)\beta \;{\bar{\;\psi_{1}\;}}^{r_{1}}\cdots \;{\bar{\;\psi_{k}\;}}^{r_{k}},\\ \psi_{1}^{r_{1}}\cdots \psi_{k}^{r_{k}} & ∼\;\bar{\;\psi_{1}^{r_{1}}\;}\cdots \;\bar{\;\psi_{k}^{r_{k}}\;},\;\mathrm{contradicting Corollary}\;2. \end{array}$$□

**Theorem** **22.**
*If an Eisenstein integer α=a+bρ and its conjugate $\;\bar{\;\alpha \;}\;$ are relatively prime, then the modular multiplicative inverse $c\equiv \alpha^{-1}\;\left(\mathrm{mod}\;\;\bar{\;\alpha \;}\;\right)$ is an integer.*


**Proof.** By Theorem 1 and recalling that $N_{\rho }\left(\alpha \right)=N_{\rho }\left(\;\bar{\;\alpha \;}\;\right)$, we have1=gcdNρ(α),23Im(α2)=gcd(a2+b2−ab,b(2a−b))=gcd(a2+b2−ab,b)=gcd(a2+b2−ab,2a−b).
Hence, there are integers *c* and *d* such that$$c\left(2a-b\right)+d\left(a^{2}+b^{2}-ab\right)=c\left(\alpha +\;\bar{\;\alpha \;}\;\right)+d\alpha \;\bar{\;\alpha \;}\;=1.$$
We verify that $c\alpha \equiv 1\;(\mathrm{mod}\;\;\bar{\;\alpha \;}\;)$ and confirm that $c\equiv \alpha^{-1}\;\left(\mathrm{mod}\;\;\bar{\;\alpha \;}\;\right)$. □

**Corollary** **3.**
*If η is an odd primitive Eisenstein integer, then the modular multiplicative inverse $c\equiv \eta^{-1}\;\left(\mathrm{mod}\;\;\bar{\;\eta \;}\;\right)$ is an integer.*


**Proof.** By Proposition 1, $\gcd(\eta ,\;\bar{\;\eta \;}\;)=1$. Applying Theorem 22 settles the claim. □

**Theorem** **23.**
*An Eisenstein integer α is an associate of $\;\bar{\;\alpha \;}\;$ if and only if α is an associate of n or kβ for some n,k∈Z.*


**Proof.** If α∼n then $\;\bar{\;\alpha \;}\;∼\;\bar{\;n\;}\;$ and n∼α. If α∼kβ for some k∈Z, then $\;\bar{\;\alpha \;}\;∼k\;\bar{\;\beta \;}\;∼k\beta ∼\alpha$. Conversely, if α=a+bρ and $\alpha ∼\;\bar{\;\alpha \;}\;$, then $\alpha =u\;\bar{\;\alpha \;}\;$ for some unit *u* in Z[ρ].
If u=1, then a+bρ=(a−b)−bρ. In this case, b=0, which implies α=a.If u=ρ, then a+bρ=b+aρ. Hence, a=b, implying α=a+aρ=a(1+ρ).If u=−(1+ρ), then a+bρ=−a+(b−a)ρ. Hence, a=0, which yields α=bρ.If u=−1, then a+bρ=(b−a)+bρ. We obtain b=2a and, hence, α=a+2aρ=a(1+2ρ).If u=−ρ, then a+bρ=−b−aρ. We obtain b=−a and, therefore, α=a−aρ=a(1−ρ).If u=1+ρ, then a+bρ=a+(a−b)ρ. We have a=2b, which means α=2b+bρ=b(2+ρ).
Having covered all cases, we confirm that α is an associate of *n* or kβ for some n,k∈Z. □

Recalling Theorem 3, we know that $\psi \neg ∼\;\bar{\;\psi \;}\;$ whenever ψ is an Eisenstein prime.

**Corollary** **4.**
*If α is a primitive Eisenstein integer such that α is not a unit and α is neither β nor any of its associates, then α and $\;\bar{\;\alpha \;}\;$ are not associates.*


**Proof.** Given the conditions on α, it is neither an associate of any n∈Z nor a multiple kβ of β with k∈Z. The conclusion follows by Theorem 23. □

### 3.2. The Group of Units as a Cyclic Group

If α is a generator element of a cyclic group *G* of order *n*, then $\alpha^{i}$ is also a generator of *G* if and only if gcd(i,n)=1. The number of generators of such a *G* is φ(n). Moreover, an α∈G is a generator of *G* if and only if $\alpha^{\frac{n}{q}}\neq 1$ for each prime divisor *q* of *n*.

The cyclic group ${\left(Z\left[\rho \right]/\langle \eta \rangle \right)}^{*}$ of order $\varphi_{\rho }\left(\eta \right)$ have $\varphi (\varphi_{\rho }\left(\eta \right))$ generators. Our next result shows that the probability of successfully selecting one generator in the cyclic group ${\left(Z\left[\rho \right]/\langle \eta \rangle \right)}^{*}$ at random is smaller than doing so in the cyclic group $Z_{n}^{*}$.

**Theorem** **24.**
*If η is an associate of some n∈N, then $\frac{\varphi (\varphi_{\rho }\left(\eta \right))}{\varphi_{\rho }\left(\eta \right)}\leq \frac{\varphi (\varphi (n))}{\varphi (n)}$.*


**Proof.** By Theorem 16, we know that $\varphi \left(n\right)|\varphi_{\rho }\left(\eta \right)$ whenever η and n∈N are associates. For a,b∈N, it is well known that, if a∣b, then $\frac{\varphi (b)}{\varphi (a)}\leq \frac{b}{a}$. Hence, we have$$\frac{\varphi (\varphi_{\rho }\left(\eta \right))}{\varphi (\varphi (n))}\leq \frac{\varphi_{\rho }\left(\eta \right)}{\varphi (n)},\;\mathrm{which implies}\;\frac{\varphi (\varphi_{\rho }\left(\eta \right))}{\varphi_{\rho }\left(\eta \right)}\leq \frac{\varphi (\varphi (n))}{\varphi (n)}.$$□

**Example** **3.**
*For the Eisenstein prime p=5, the order of the cyclic group ${\left(Z\left[\rho \right]/\langle 5\rangle \right)}^{*}≅{\left(Z_{5}\left[\rho \right]\right)}^{*}$ is $\varphi_{\rho }\left(5\right)=5^{2}-1=24$ whose prime factorization is $\varphi_{\rho }\left(5\right)=2^{3}\cdot 3$. Let α be a generator of ${\left(Z_{5}\left[\rho \right]\right)}^{*}$. It suffices to show that*

$$\alpha^{\frac{\varphi_{\rho }\left(5\right)}{3}}=\alpha^{8}\neq 1\;\left(\mathrm{mod}\;5\right)\;and\;\alpha^{\frac{\varphi_{\rho }\left(5\right)}{2}}=\alpha^{12}\neq 1\;\left(\mathrm{mod}\;5\right).$$

*We can select α:=2+ρ to generate ${\left(Z_{5}\left[\rho \right]\right)}^{*}$, since $\alpha^{8}=4+4\rho \;\left(\mathrm{mod}\;5\right)$ and $\alpha^{12}=4\;\left(\mathrm{mod}\;5\right)$. The other seven generators are*

$$\begin{array}{cccccccc} \alpha^{5} & =1+4\rho , & \alpha^{7} & =1+3\rho , & \alpha^{11} & =3+2\rho , & \alpha^{13} & =3+4\rho ,\\ \alpha^{17} & =4+\rho , & \alpha^{19} & =4+2\rho , & \alpha^{23} & =2+3\rho . \end{array}$$

*The group $Z_{5}^{*}$ has φ(φ(5))=φ(4)=2 generators, namely, 2 and 3. It is clear that*

$$\frac{\varphi (\varphi_{\rho }\left(5\right))}{\varphi_{\rho }\left(5\right)}=\frac{8}{24}<\frac{2}{4}=\frac{\varphi (\varphi (5))}{\varphi (5)}.$$



**Theorem** **25.**
*If ${\left(Z\left[\rho \right]/\langle \eta \rangle \right)}^{*}$ is a cyclic group, then*

$$\prod_{\alpha \in {\left(Z\left[\rho \right]/\langle \eta \rangle \right)}^{*}}\alpha \equiv -1\;\left(\mathrm{mod}\;\eta \right).$$



**Proof.** Let ${\left(Z\left[\rho \right]/\langle \eta \rangle \right)}^{*}$ be a cyclic group. By Theorem 13, η is an element in the set $\left\{\beta ,\beta^{2},2\beta ,\psi^{k}\right\}$, a prime p≡2(mod3), or any of their associates. We investigate by the values that η takes.If γ=β, then, by Theorem 12, we get ${\left(Z\left[\rho \right]/\langle \beta \rangle \right)}^{*}=\left\{{\left[1\right]}_{\beta },{\left[2\right]}_{\beta }\right\}$. Hence,$$\prod_{\alpha \in {\left(Z\left[\rho \right]/\langle \beta \rangle \right)}^{*}}\alpha \equiv 1\cdot 2\equiv 2\equiv -1\;\left(\mathrm{mod}\;\beta \right).$$If γ=2, then ${\left(Z\left[\rho \right]/\langle 2\rangle \right)}^{*}$ is a cyclic group of order $\varphi_{\rho }\left(2\right)=3$. Letting θ be a generator,$$\prod_{\alpha \in {\left(Z\left[\rho \right]/\langle 2\rangle \right)}^{*}}\alpha \equiv \prod_{0\leq t\leq 2}\theta^{t}\equiv \theta^{3}\;\left(\mathrm{mod}\;2\right).$$
Since the generators of ${\left(Z\left[\rho \right]/\langle 2\rangle \right)}^{*}$ are ρ and 1+ρ, we have $\theta^{3}\equiv 1\equiv -1\;\left(\mathrm{mod}\;2\right)$.If $\gamma \in \{\beta^{2},2\beta ,\psi^{k}\}$ or γ=p≡2(mod3) such that p≠2, then $\varphi_{\rho }\left(\gamma \right)$ is an even number, by Theorem 14. Letting θ be a generator of ${\left(Z\left[\rho \right]/\langle \gamma \rangle \right)}^{*}$,$$\prod_{\alpha \in {\left(Z\left[\rho \right]/\langle \gamma \rangle \right)}^{*}}\alpha \equiv \prod_{0\leq t\leq \varphi_{\rho }\left(\gamma \right)-1}\theta^{t}\equiv \theta^{\frac{\varphi_{\rho }\left(\gamma \right)\left(\varphi_{\rho }\left(\gamma \right)-1\right)}{2}}\;\left(\mathrm{mod}\;\gamma \right).$$
The order of θ is an even number $\varphi_{\rho }\left(\gamma \right)$. Hence, $\theta^{\frac{\varphi_{\rho }\left(\gamma \right)}{2}}\in {\left(Z\left[\rho \right]/\langle \gamma \rangle \right)}^{*}$ must be −1 because −1 is the only element of order 2 in ${\left(Z\left[\rho \right]/\langle \gamma \rangle \right)}^{*}$. Since $\varphi_{\rho }\left(\gamma \right)-1$ is an odd number,$$\theta^{\frac{\varphi_{\rho }\left(\gamma \right)\left(\varphi_{\rho }\left(\gamma \right)-1\right)}{2}}={\left(\theta^{\frac{\varphi_{\rho }\left(\gamma \right)}{2}}\right)}^{\varphi_{\rho }\left(\gamma \right)-1}\;\left(\mathrm{mod}\;\gamma \right)={\left(-1\right)}^{\varphi_{\rho }\left(\gamma \right)-1}\;\left(\mathrm{mod}\;\gamma \right)=-1\;\left(\mathrm{mod}\;\eta \right).$$□

The Wilson Theorem over Z had been generalized to Gaussian integers in [35], but not to Eisenstein integers in the prior literature. We highlight that we have achieved this as a special case of Theorem 25.

**Theorem** **26**(Wilson Theorem for Eisenstein Integers). *If γ∈Z[ρ] is an Eisenstein prime, then*$$\prod_{\alpha \in {\left(Z\left[\rho \right]/\langle \gamma \rangle \right)}^{*}}\alpha \equiv -1\;\left(\mathrm{mod}\;\gamma \right).$$

## 4. Set Partitioning Based on the Multiplicative Group

In a recent work [8], we proposed a number of Eisenstein constellations $E_{\eta }$ as two-dimensional signal constellations by using the modulo function in (Equation 1). The setup, given a suitable η, has(3)$$\begin{array}{cc} E_{\eta } & =\{\mu_{\eta }\left(\alpha \right)\;:\;\alpha \in R_{\eta }\}\;\mathrm{with}\\ R_{\eta } & =\{x+y\rho \;:\;0\leq x<tN_{\rho }\left(m+n\rho \right)\;\mathrm{and}\;0\leq y<t\}. \end{array}$$

In that work, we also introduced set partitioning of Eisenstein integers based on *additive* subgroups. In this section, we focus on set partitioning based on the *multiplicative* group.

We now propose Eisenstein constellations ${\left(E_{\eta }\right)}^{*}$, corresponding to the cyclic group ${\left(Z\left[\rho \right]/\langle \eta \rangle \right)}^{*}$, with $\eta \in \{\beta^{2},2\beta ,\psi^{k}\;:\;k\in N\}$ or η being an odd prime integer p≡2(mod3). In doing this, we generalize Proposition 1 in [18], which covers the case of η=ψ. Our set partitioning technique for signal constellation ${\left(E_{\eta }\right)}^{*}$ benefits from the facts that ${\left(Z\left[\rho \right]/\langle \eta \rangle \right)}^{*}$ is a cyclic group of order $\varphi_{\rho }\left(\eta \right)$, by Theorem 13, and $\varphi_{\rho }\left(\eta \right)\equiv 0\;\left(\mathrm{mod}\;6\right)$, by Theorem 15. The elements of ${\left(E_{\eta }\right)}^{*}$ can be expressed as powers of a generator α as$${\left(E_{\eta }\right)}^{*}=\left\{\alpha^{0},\alpha^{1},\ldots ,\alpha^{\varphi_{\rho }\left(\eta \right)-1}\right\}.$$
Letting $n:=\frac{\varphi_{\rho }\left(\eta \right)}{6}$, the set of all unit (see [29] for η=ψ) is$$\left\{\alpha^{n},\alpha^{2n},\alpha^{3n},\alpha^{4n},\alpha^{5n},\alpha^{6n}\right\}=\left\{\pm 1,\pm \rho ,\pm \left(1+\rho \right)\right\}.$$
We can then partition ${\left(E_{\eta }\right)}^{*}$ into *n* subsets, indexed by j∈{0,1,…,n−1}, as$${\left(E_{\eta }\right)}_{\left(j\right)}^{*}=\left\{\alpha^{n+j},\alpha^{2n+j},\alpha^{3n+j},\alpha^{4n+j},\alpha^{5n+j},\alpha^{6n+j}\right\}=\left\{\pm \alpha^{j},\pm \rho \alpha^{j},\pm \left(1+\rho \right)\alpha^{j}\right\}.$$
All elements of ${\left(E_{\eta }\right)}^{*}$ can be found by calculating $\alpha^{j}$ for j∈{0,1,…,n−1} using the modulo function in (Equation 1), followed by multiplying each $\alpha^{j}$ by the units.

Our next result extends Theorem 2 to the cases $\eta \in \{2\beta ,\beta^{2},\psi^{k}\;:\;k\in N\}$ or η being an odd prime integer p≡2(mod3).

**Theorem** **27.**
*Let $\eta \in \{2\beta ,\beta^{2},\psi^{k}\;:\;k\in N\}$ or η=p, with p≡2(mod3) being an odd prime. If α is a generator of ${\left(E\right)}_{\eta }^{*}$, then the minimum Euclidean distance in the subset ${\left(E_{\eta }\right)}_{\left(j\right)}^{*}$ is $∥\alpha^{j}∥$. Furthermore, ${\left(E_{\eta }\right)}_{\left(j\right)}^{*}$ can be partitioned into three subsets*

$${\left(E_{\eta }\right)}_{\left(j\right)}^{*}=\left\{\pm \alpha^{j}\right\}\cup \left\{\pm \rho \alpha^{j}\right\}\cup \left\{\pm \left(1+\rho \right)\alpha^{j}\right\},$$

*each with minimum Euclidean distance $2∥\alpha^{j}∥$. We also can partition ${\left(E_{\eta }\right)}_{\left(j\right)}^{*}$ into two subsets*

$${\left(E_{\eta }\right)}_{\left(j\right)}^{*}=\left\{\alpha^{j},\rho \alpha^{j},-\left(1+\rho \right)\alpha^{j}\right\}\cup \left\{-\alpha^{j},-\rho \alpha^{j},\left(1+\rho \right)\alpha^{j}\right\},$$

*each with minimum Euclidean distance $\sqrt{3}∥\alpha^{j}∥$.*


**Proof.** Two neighboring points in ${\left(E_{\eta }\right)}_{\left(j\right)}^{*}$ have a phase difference of π/3. Hence, the pair together with the origin form an equilateral triangle whose sides are of length $∥\alpha^{j}∥$, confirming that the minimum Euclidean distance is $∥\alpha^{j}∥$.The sets $\left\{\pm \alpha^{j}\right\},\left\{\pm \rho \alpha^{j}\right\}$ and $\left\{\pm \left(1+\rho \right)\alpha^{j}\right\}$ contain points whose pairwise phase difference is π, ensuring the minimum distance $2∥\alpha^{j}∥$. The sets $\left\{\alpha^{j},\rho \alpha^{j},-\left(1+\rho \right)\alpha^{j}\right\}$ and $\left\{-\alpha^{j},-\rho \alpha^{j},\left(1+\rho \right)\alpha^{j}\right\}$ contain points whose pairwise phase difference is 2π/3, yielding the minimum distance of $\sqrt{3}∥\alpha^{j}∥$. □

**Example** **4.**
*(Primitive but not prime) Given a primitive Eisenstein $\psi^{2}=-5+3\rho$, with ψ=2+3ρ, we have the cyclic group ${\left(Z\left[\rho \right]/\langle -5+3\rho \rangle \right)}^{*}≅{\left(E_{-5+3\rho }\right)}^{*}$ generated by α=3. Since $\varphi_{\rho }\left(\psi^{2}\right)=42$, we can partition ${\left(E_{\psi^{2}}\right)}^{*}$ into 7 subsets defined as*

$${\left(E_{\psi^{2}}\right)}_{\left(j\right)}^{*}=\left\{\pm \alpha^{j},\pm \rho \alpha^{j},\pm \left(1+\rho \right)\alpha^{j}\right\}with\;j\in \left\{0,1,2,3,4,5,6\right\}.$$

*Since α=3, by using the modulo function (Equation 1), we have*

$$\alpha^{2}=-4-2\rho ,\;\alpha^{3}=-4-\rho ,\;\alpha^{4}=1-\rho ,\;\alpha^{5}=-2,\;\alpha^{6}=-1-3\rho .$$

*We rely on Theorem 27 to partition ${\left(E_{\psi^{2}}\right)}_{\left(j\right)}^{*}$ into three subsets and two subsets, each with respective minimum Euclidean distances $2∥\alpha^{j}∥$ and $\sqrt{3}∥\alpha^{j}∥$ for j∈{0,1,2,3,4,5,6} as follows:*

$$\begin{array}{cc} {\left(E_{\psi^{2}}\right)}_{\left(0\right)}^{*} & =\{1,\;\rho ,\;1+\rho ,\;-1,-\rho ,\;-1-\rho \},\\ \; & =\{1,\;-1\}\cup \{\rho ,\;-\rho \}\cup \{-1-\rho ,\;1+\rho \},\\ \; & =\{1,\;\rho ,\;-1-\rho \}\cup \{-1,-\rho ,1+\rho \},\\ {\left(E_{\psi^{2}}\right)}_{\left(1\right)}^{*} & =\{3,\;3+3\rho ,\;3\rho ,\;-3,\;-3-3\rho ,\;-3\rho \},\\ \; & =\{3,\;-3\}\cup \{3\rho ,\;-3\rho \}\cup \{-3-3\rho ,\;3+3\rho \},\\ \; & =\{3,\;3\rho ,\;-3-3\rho \}\cup \{-3,\;-3\rho ,\;3+3\rho \},\\ {\left(E_{\psi^{2}}\right)}_{\left(2\right)}^{*} & =\{4+2\rho ,\;2+4\rho ,\;-2+2\rho ,\;-4-2\rho ,\;-2-4\rho ,\;2-2\rho \},\\ \; & =\{4+2\rho ,\;-4-2\rho \}\cup \{2+4\rho ,\;-2-4\rho \}\cup \{-2+2\rho ,\;2-2\rho \},\\ \; & =\{4+2\rho ,\;-2-4\rho \;-2+2\rho \}\cup \{\;-4-2\rho ,\;2+4\rho ,\;2-2\rho \},\\ {\left(E_{\psi^{2}}\right)}_{\left(3\right)}^{*} & =\{4+\rho ,\;3+4\rho ,\;-1+3\rho ,\;-4-\rho ,\;-3-4\rho ,\;1-3\rho \},\\ \; & =\{4+\rho ,\;-4-\rho \}\cup \{3+4\rho ,\;-3-4\rho \}\cup \{-1+3\rho ,\;1-3\rho \},\\ \; & =\{4+\rho ,\;-1+3\rho ,\;-3-4\rho \}\cup \{-4-\rho ,,\;1-3\rho ,\;3+4\rho \},\\ {\left(E_{\psi^{2}}\right)}_{\left(4\right)}^{*} & =\{2+\rho ,\;1+2\rho ,\;-1+\rho ,\;-2-\rho ,\;-1-2\rho ,\;1-\rho \},\\ \; & =\{2+\rho ,\;-2-\rho \}\cup \{-1+\rho ,\;1-\rho \}\cup \{-1-2\rho ,\;1+2\rho \},\\ \; & =\{2+\rho ,\;-1-2\rho ,\;-1+\rho \}\cup \{-2-\rho ,\;1+2\rho ,\;1-\rho \},\\ {\left(E_{\psi^{2}}\right)}_{\left(5\right)}^{*} & =\{2,\;2+2\rho ,\;2\rho ,\;-2,\;-2-2\rho ,\;-2\rho \},\\ \; & =\{2,\;-2\}\cup \{2\rho ,\;-2\rho \}\cup \{-2-2\rho ,\;2+2\rho \},\\ \; & =\{2,\;2\rho ,\;-2-2\rho ,\}\cup \{-2,\;2+2\rho ,\;-2\rho \},\\ {\left(E_{\psi^{2}}\right)}_{\left(6\right)}^{*} & =\{3+2\rho ,\;1+3\rho ,\;-2+\rho ,\;-3-2\rho ,\;-1-3\rho ,\;2-\rho \},\\ \; & =\{3+2\rho ,-3-2\rho \}\cup \{1+3\rho ,\;-1-3\rho \}\cup \{-2+\rho ,\;2-\rho \},\\ \; & =\{3+2\rho ,\;-1-3\rho ,\;-2+\rho \}\cup \{-3-2\rho ,\;1+3\rho ,\;2-\rho \}. \end{array}$$


*Figure 1, Figure 2 and Figure 3 visualize the Eisenstein constellations ${\left(E_{\psi^{2}}\right)}^{*}$ and its signal partitions in C.*


**Example** **5.**
*(Prime but not primitive) Given an Eisenstein prime p=5, we have the cyclic group ${\left(Z\left[\rho \right]/\langle 5\rangle \right)}^{*}≅{\left(E_{5}\right)}^{*}$ generated by α=2+ρ. Since $\varphi_{\rho }\left(5\right)=24$, we can partition ${\left(E_{5}\right)}^{*}$ into 4 subsets as*

$${\left(E_{5}\right)}_{\left(j\right)}^{*}=\left\{\pm \alpha^{j},\pm \rho \alpha^{j},\pm \left(1+\rho \right)\alpha^{j}\right\},with\;j\in \left\{0,1,2,3\right\}.$$

*Since α=2+ρ, the modulo function in (Equation 1) gives us $\alpha^{2}=-2-2\rho$ and $\alpha^{3}=-2+\rho$. By Theorem 27, we partition ${\left(E_{5}\right)}_{\left(j\right)}^{*}$ into three and two subsets each with respective minimum Euclidean distances $2∥\alpha^{j}∥$ and $\sqrt{3}∥\alpha^{j}∥$ for j∈{0,1,2,3} as follows:*

$$\begin{array}{cc} {\left(E_{5}\right)}_{\left(0\right)}^{*} & =\{1,\;\rho ,\;1+\rho ,\;-1,-\rho ,\;-1-\rho \},\\ \; & =\{1,\;-1\}\cup \{\rho ,\;-\rho \}\cup \{-1-\rho ,\;1+\rho \},\\ \; & =\{1,\;\rho ,\;-1-\rho \}\cup \{-1,-\rho ,1+\rho \},\\ {\left(E_{5}\right)}_{\left(1\right)}^{*} & =\{2+\rho ,\;1+2\rho ,\;-1+\rho ,\;-2-\rho ,\;-1-2\rho ,\;1-\rho \},\\ \; & =\{2+\rho ,\;-2-\rho \}\cup \{-1+\rho ,\;1-\rho \}\cup \{-1-2\rho ,\;1+2\rho \},\\ \; & =\{2+\rho ,\;-1-2\rho ,\;-1+\rho \}\cup \{-2-\rho ,\;1+2\rho ,\;1-\rho \},\\ {\left(E_{5}\right)}_{\left(2\right)}^{*} & =\{2,\;2+2\rho ,\;2\rho ,\;-2,\;-2-2\rho ,\;-2\rho \},\\ \; & =\{2,\;-2\}\cup \{2\rho ,\;-2\rho \}\cup \{-2-2\rho ,\;2+2\rho \},\\ \; & =\{2,\;-2-2\rho ,\;2\rho \}\cup \{-2,\;2+2\rho ,\;-2\rho \},\\ {\left(E_{5}\right)}_{\left(3\right)}^{*} & =\{3+2\rho ,\;-3-2\rho ,\;-2+\rho ,\;2-\rho ,\;-1-3\rho ,\;1+3\rho \},\\ \; & =\{3+2\rho ,\;-3-2\rho \}\cup \{-2+\rho ,\;2-\rho \}\cup \{-1-3\rho ,\;1+3\rho \},\\ \; & =\{3+2\rho ,\;-1-3\rho ,\;-2+\rho \}\cup \{-3-2\rho ,\;2-\rho ,\;1+3\rho \}. \end{array}$$


*Figure 4, Figure 5 and Figure 6 visualize the constellation ${\left(E_{5}\right)}^{*}$ and its signal partitions in C.*


## 5. Discussion

We can use a primitive Eisenstein integer η to construct signal constellations and complex-valued codes over Eisenstein integers. We consider the sets $R_{\eta }$ and $E_{\eta }$ in (Equation 3) as code alphabets, where $E_{\eta }$ is obtained through the modulo function in (Equation 1), based on an isomorphism between Z[ρ] modulo a primitive Eisenstein integer η and Z modulo a norm of the primitive Eisenstein as in Theorem 8. Codes over Eisenstein integers whose alphabet set $E_{\psi }$ is an Eisenstein field of cardinality a prime q≡1(mod3) were investigated in [29] and [36]. The Eisenstein field corresponds to a quotient ring of Eisenstein integers over an ideal generated by a prime and a primitive Eisenstein integer ψ. More generally, a recent code construction via a quotient ring of Eisenstein integers induced by an ideal generated by a primitive but not a prime Eisenstein integer can be found in [8]. Table 1 provides an example. The alphabet set $E_{\psi^{2}}$ is obtained from the quotient ring $Z\left[\rho \right]/\langle \psi^{2}\rangle$ with primitive Eisenstein $\psi^{2}=-5+3\rho$ and ψ=2+3ρ via the modulo function in (Equation 1).

A *code* is a nonempty subset $C\subseteq E_{\eta }^{n}$ whose elements are called *codewords*. A *linear codeC* of length *n* over $E_{\eta }$ is a submodule of $E_{\eta }^{n}$. Since $E_{\eta }$ and $E_{\eta }^{n}$ are abelian groups, we say that *C* is a *group code* if it is a subgroup of $E_{\eta }^{n}$. When $E_{\eta }$ is a finite field, that is, $E_{\eta }^{n}$ is a vector space of dimension *n* over $E_{\eta }$, a linear code *C* is a subspace of $E_{\eta }^{n}$. We call *C* an (n,k) code if *C* has exactly $|E_{\eta }{|}^{k}$ codewords.

By Corollaries 2 and 4, odd primitives η and $\;\bar{\;\eta \;}\;$ are not associates. Hence, $\langle \eta \rangle \neq \langle \;\bar{\;\eta \;}\;\rangle$ and, therefore, $Z\left[\rho \right]/\langle \eta \rangle \neq Z\left[\rho \right]/\langle \;\bar{\;\eta \;}\;\rangle$. By Proposition 1, odd primitives $\psi_{1}^{r_{1}}\cdots \psi_{k}^{r_{k}}$ and $\;\bar{\;\psi_{1}^{r_{1}}\cdots \psi_{k}^{r_{k}}\;}\;$ are relatively prime. By the Chinese Remainder Theorem (CRT) with $N_{\rho }\left(\psi_{i}\right)=q_{i}$ being a prime integer such that $q_{i}\equiv 1\;\left(\mathrm{mod}\;3\right)$, we have$$Z\left[\rho \right]/\langle q_{1}^{r_{1}}\cdots q_{k}^{r_{k}}\rangle ≅Z\left[\rho \right]/\langle \psi_{1}^{r_{1}}\cdots \psi_{k}^{r_{k}}\rangle \times Z\left[\rho \right]/\langle \;\bar{\;\psi_{1}^{r_{1}}\cdots \psi_{k}^{r_{k}}\;}\;\rangle .$$

For an even primitive Eisenstein integer η, however, the CRT does not hold. Hence,$$Z\left[\rho \right]/\langle n\rangle \neg ≅Z\left[\rho \right]/\langle \eta \rangle \times Z\left[\rho \right]/\langle \;\bar{\;\eta \;}\;\rangle withN_{\rho }\left(\eta \right)=n.$$

Set partitioning based on an *additive* subgroup is structurally *not feasible* on the Eisenstein field $E_{\psi }$ due to its cardinality being a prime integer. Hence, set partitioning based on a *multiplicative* group of the Eisenstein field $E_{\psi }$ was proposed in [18]. The investigation leveraged on the fact that a multiplicative group of the Eisenstein field is cyclic to perform set partitioning. Theorem 27 is an insightful generalization. It extends set partitioning to a multiplicative group of a quotient ring of Eisenstein integers when the group is designed to be cyclic.

Given a primitive Eisenstein integer η, the quotient ring $Z\left[\rho \right]/\langle \eta \rangle ≅Z_{N_{\rho }\left(\eta \right)}$ defines a finite set of representative elements that form the signal constellation$$E_{\eta }=\left\{\mu_{\eta }\left(\alpha \right):\alpha \in Z_{N_{\rho }\left(\eta \right)}\right\}.$$

This constitutes a special case of (Equation 3), where $\mu_{\eta }\left(\alpha \right)$ denotes the modulo function in (Equation 1) applied to an Eisenstein integer α. Such a structure is fundamental in designing multidimensional lattice codes. It enables efficient encoding and decoding procedures. By integrating Eisenstein constellations into coding theory, we establish a direct link between complex-valued codes and structured lattice-based signal constellations. The resulting codes benefit from increased minimum Euclidean distances, enhancing signal robustness in noisy communication channels.

## 6. Summary and Concluding Remarks

We have just reported properties of primitive, even, or odd Eisenstein integers. For the odd ones, we investigated whether they are of Type 1 or 2 and their implied properties according to the type.

Given an Eisenstein prime ψ such that $N_{\rho }\left(\psi \right)=q$ is a prime integer equivalent to 1(mod3), we settled the question posed as Question 6.1 in [7]. If ψ and $\;\bar{\;\psi \;}\;$ are distinct Eisenstein primes which are not associates, then they belong to the same odd class. If one of them is of Type 1, then the other is also of Type 1. The same goes for Type 2. The corresponding *q*, however, is insufficient to conclude which odd class ψ and $\;\bar{\;\psi \;}\;$ belong to.

We have confirmed that, if Eisenstein integers α and $\;\bar{\;\alpha \;}\;$ are relatively prime, then $\alpha^{-1}\;\left(\mathrm{mod}\;\;\bar{\;\alpha \;}\;\right)$ is in Z. We also managed to prove that the multiplicative group of the set of all units in a quotient ring of Z[ρ] forms a cyclic group. This leads to a nice set partitioning, allowing us to propose Eisenstein signal constellations. Some examples were given to further illustrate the insights.

Many algebraic signal constellations have been known to enhance the performance of communication systems. Studying use cases and measuring the optimality of certain families of constellations form an important topic in modern communications. Constructing good constellations and benchmarking their performance against previously best-known ones, either in general or for specific setups, are interesting directions to consider.

## Figures and Tables

**Figure 1 entropy-27-00337-f001:**
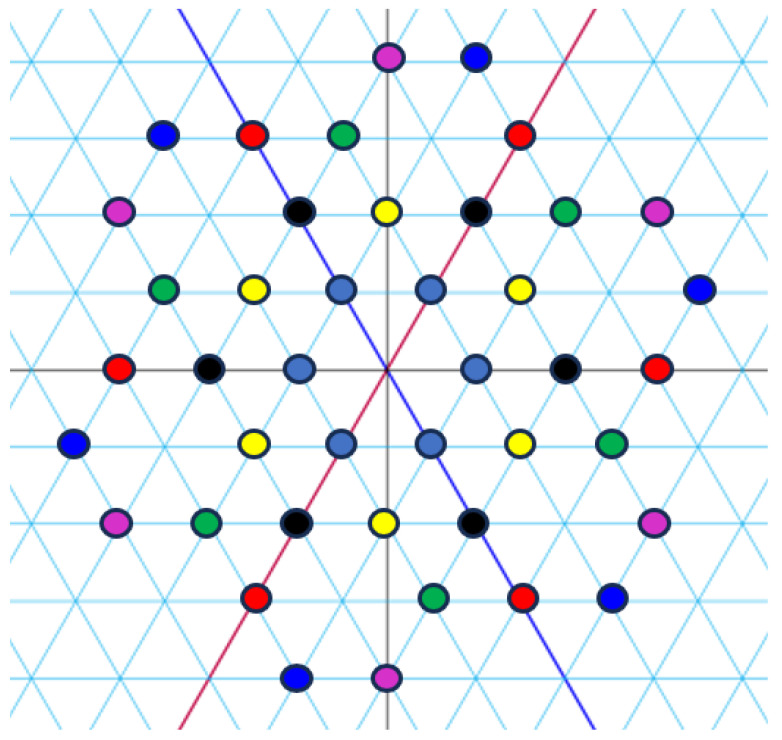
Set partitioning of ${\left(E_{\psi^{2}}\right)}^{*}$ into seven subsets. Circles represent the integers, with colours corresponding to indices.

**Figure 2 entropy-27-00337-f002:**
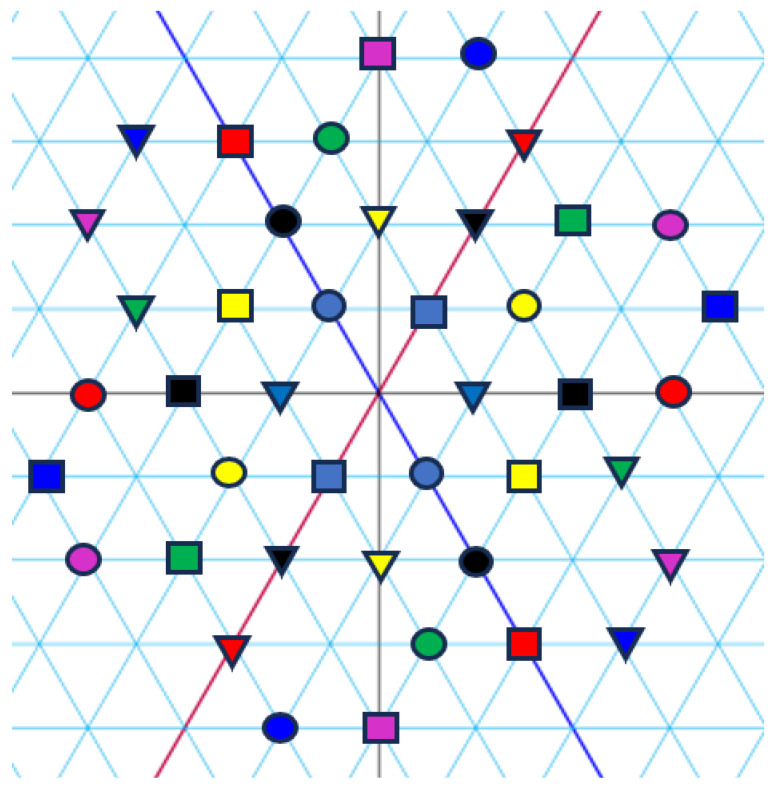
Set partitioning of ${\left(E_{\psi^{2}}\right)}_{\left(j\right)}^{*}$ into three subsets. Colours correspond to indices. Forms (circle, square, and triangle) correspond to subsets.

**Figure 3 entropy-27-00337-f003:**
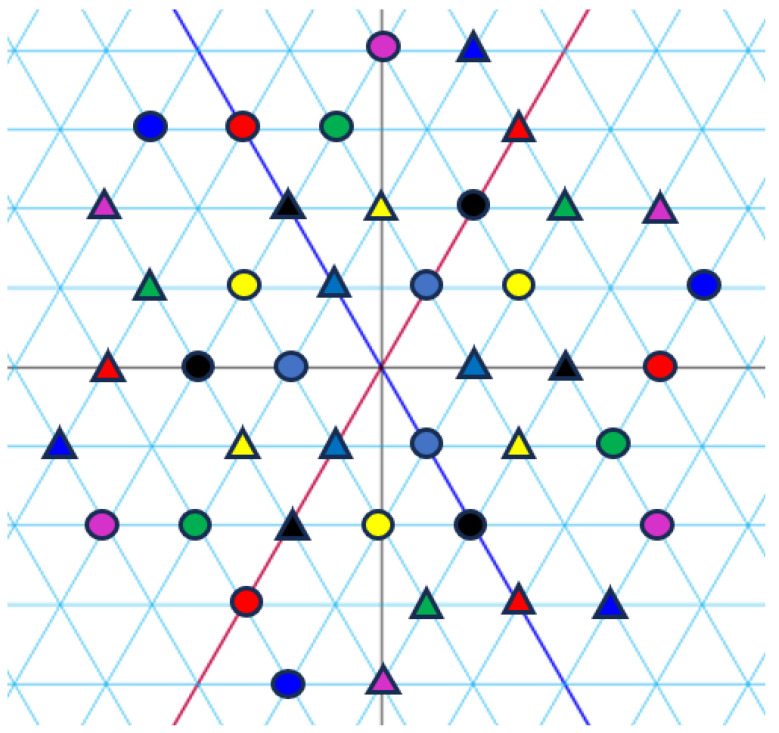
Set partitioning of ${\left(E_{\psi^{2}}\right)}_{\left(j\right)}^{*}$ into two subsets. Colours correspond to indices. Forms (circle and triangle) correspond to subsets.

**Figure 4 entropy-27-00337-f004:**
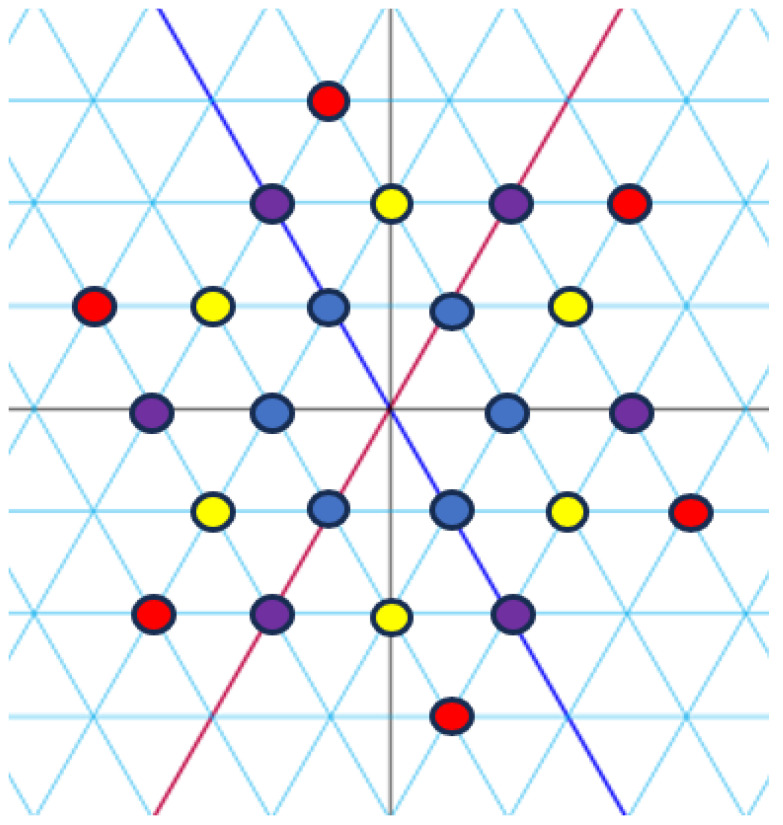
Set partitioning of ${\left(E_{5}\right)}^{*}$ into four subsets. Circles represent the integers, with colours corresponding to indices.

**Figure 5 entropy-27-00337-f005:**
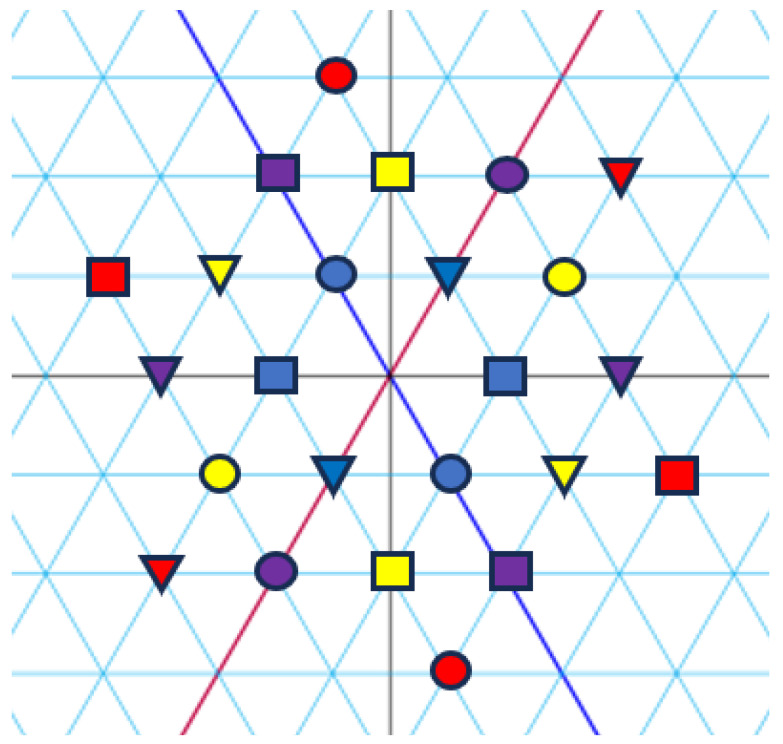
Set partitioning of ${\left(E_{5}\right)}_{\left(j\right)}^{*}$ into three subsets. Forms (circle, square, and triangle) correspond to subsets. Colours correspond to indices.

**Figure 6 entropy-27-00337-f006:**
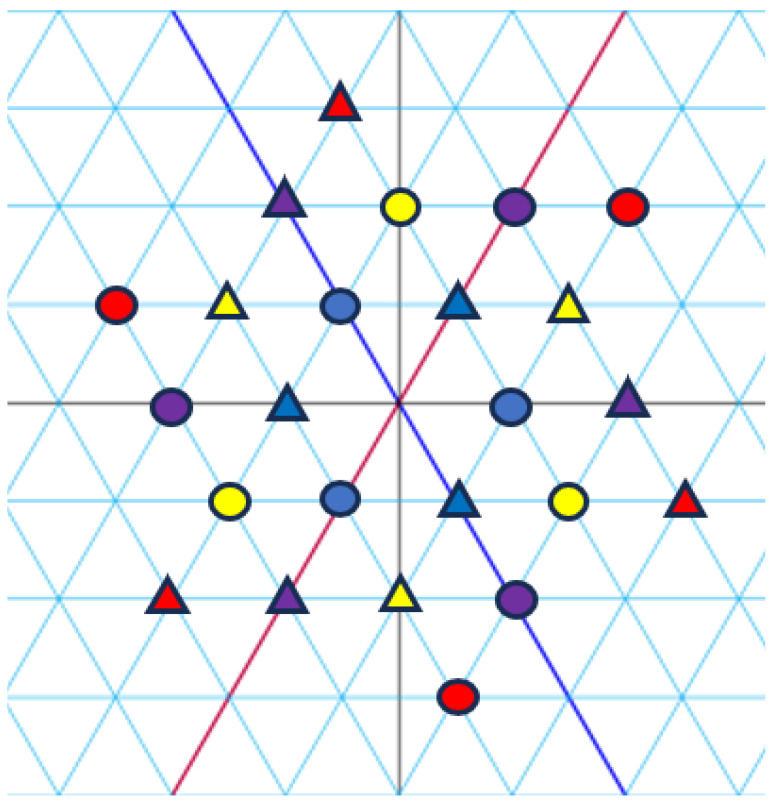
Set partitioning of ${\left(E_{5}\right)}_{\left(j\right)}^{*}$ into two subsets. Form (circle and triangle) correspond to subsets. Colours correspond to indices.

**Table 1 entropy-27-00337-t001:** Elements in $Z\left[\rho \right]/\langle \psi^{2}\rangle ≅Z_{49}$ and $E_{\psi^{2}}$.

$Z\left[\rho \right]/\langle \psi^{2}\rangle$	$E_{\psi^{2}}$	$Z\left[\rho \right]/\langle \psi^{2}\rangle$	$E_{\psi^{2}}$	$Z\left[\rho \right]/\langle \psi^{2}\rangle$	$E_{\psi^{2}}$
${\left[0\right]}_{\psi^{2}}$	0	${\left[17\right]}_{\psi^{2}}$	−1+ρ	${\left[34\right]}_{\psi^{2}}$	−2+2ρ
${\left[1\right]}_{\psi^{2}}$	1	${\left[18\right]}_{\psi^{2}}$	ρ	${\left[35\right]}_{\psi^{2}}$	−1+2ρ
${\left[2\right]}_{\psi^{2}}$	2	${\left[19\right]}_{\psi^{2}}$	1+ρ	${\left[36\right]}_{\psi^{2}}$	2ρ
${\left[3\right]}_{\psi^{2}}$	3	${\left[20\right]}_{\psi^{2}}$	2+ρ	${\left[37\right]}_{\psi^{2}}$	1+2ρ
${\left[4\right]}_{\psi^{2}}$	−1+3ρ	${\left[21\right]}_{\psi^{2}}$	3+ρ	${\left[38\right]}_{\psi^{2}}$	2+2ρ
${\left[5\right]}_{\psi^{2}}$	3ρ	${\left[22\right]}_{\psi^{2}}$	4+ρ	${\left[39\right]}_{\psi^{2}}$	3+2ρ
${\left[6\right]}_{\psi^{2}}$	1+3ρ	${\left[23\right]}_{\psi^{2}}$	−3−4ρ	${\left[40\right]}_{\psi^{2}}$	4+2ρ
${\left[7\right]}_{\psi^{2}}$	2+3ρ	${\left[24\right]}_{\psi^{2}}$	−2−4ρ	${\left[41\right]}_{\psi^{2}}$	−3−3ρ
${\left[8\right]}_{\psi^{2}}$	3+3ρ	${\left[25\right]}_{\psi^{2}}$	2+4ρ	${\left[42\right]}_{\psi^{2}}$	−2−3ρ
${\left[9\right]}_{\psi^{2}}$	−4−2ρ	${\left[26\right]}_{\psi^{2}}$	3+4ρ	${\left[43\right]}_{\psi^{2}}$	−1−3ρ
${\left[10\right]}_{\psi^{2}}$	−3−2ρ	${\left[27\right]}_{\psi^{2}}$	−4−ρ	${\left[44\right]}_{\psi^{2}}$	−3ρ
${\left[11\right]}_{\psi^{2}}$	−2−2ρ	${\left[28\right]}_{\psi^{2}}$	−3−ρ	${\left[45\right]}_{\psi^{2}}$	1−3ρ
${\left[12\right]}_{\psi^{2}}$	−1−2ρ	${\left[29\right]}_{\psi^{2}}$	−2−ρ	${\left[46\right]}_{\psi^{2}}$	−3
${\left[13\right]}_{\psi^{2}}$	−2ρ	${\left[30\right]}_{\psi^{2}}$	−1−ρ	${\left[47\right]}_{\psi^{2}}$	−2
${\left[14\right]}_{\psi^{2}}$	1−2ρ	${\left[31\right]}_{\psi^{2}}$	−ρ	${\left[48\right]}_{\psi^{2}}$	−1
${\left[15\right]}_{\psi^{2}}$	2−2ρ	${\left[32\right]}_{\psi^{2}}$	1−ρ		
${\left[16\right]}_{\psi^{2}}$	−2+ρ	${\left[33\right]}_{\psi^{2}}$	2−ρ		

## Data Availability

The data is contained within the article.

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
