# Peer review of "On Algebraic Properties of Primitive Eisenstein Integers with Applications in Coding Theory"

_entropy, 2025, doi:10.3390/e27040337_

Round 1
Reviewer 1 Report
Comments and Suggestions for Authors
The paper is well written, but I have two comments:
- The title deals with applications in coding theory, and the authors consider signal constellations for the general spatial modulation. They propose Eisenstein constellations with special parameters. But they don't say anything about complex-valued codes other than references to various papers in the introduction. I recommend that the authors include a definition of these codes and also their relationship to signal constellations.
- The Eisenstein integers are sometimes also called the Eisenstein-Jacobi integers (Wolfram MathWorld) or Eulerian integers (Wikipedia), etc. Please, write a little bit more about the history of these complex numbers.
Author Response
Our reply to Reviewer 1 is in the attached pdf file.

Reviewer 2 Report
Comments and Suggestions for Authors
Dear authors, The manuscript is well-written, and the results are interesting (examples 4 and 5 explain the developed topics well). Please consider the following minor revisions to improve it:
- Some misprints: For an Eisenstein integers...;
- Row 8, page 1, needs a reference.
- Row 17, page 2, needs a reference.
- Theorems 12-14, should be referenced with its corresponding number.
- Theorem 18 is a Remark.
- Theorems 21, 27, and 27 should be in the preliminaries section.
- The relationship between the obtained results and the coding theory is not clear.
Author Response
Please see the attached pdf file for our reply.
